# Evaluating the feasibility and acceptability of an exercise and behaviour change intervention in socioeconomically deprived patients with peripheral arterial disease: The textpad study protocol

**Gabriel Cucato**[1]*, **Chris Snowden**[2], **Emma McCone**[3], **Craig Nesbitt**[3], **Sandip Nandhra**[3], **Mackenzie Fong**[4], **Eileen Kane**[4], **Maisie Rowland**[4], **Nawaraj Bhattarai**[4], **Paul Court**[5], **Oliver Bell**[6], **John Michael Saxton**[7], **James Prentis**[2]

1 Dept of Sport, Exercise and Rehabilitation, Northumbria University, Newcastle upon Tyne, United Kingdom, 2 Dept of Perioperative and Critical Care Medicine, Freeman Hospital, Newcastle upon Tyne, United Kingdom, 3 Northern Vascular Unit, Freeman Hospital, Newcastle upon Tyne, United Kingdom, 4 Newcastle University, Newcastle upon Tyne, United Kingdom, 5 Healthworks, Newcastle upon Tyne, United Kingdom, 6 Newcastle United Foundation, Newcastle upon Tyne, United Kingdom, 7 Dept of Sport, Health & Exercise Science, University of Hull, Hull, United Kingdom

* gacucato@gmail.com

**Data Availability Statement:** No datasets were generated or analysed during the current study. All

## Abstract

This pilot randomised controlled trial aims to assess the feasibility and acceptability of a 12-week home-based telehealth exercise and behavioural intervention delivered in socioeconomically deprived patients with peripheral artery disease (PAD). The study will also determine the preliminary effectiveness of the intervention for improving clinical and health outcomes. Sixty patients with PAD who meet the inclusion criteria will be recruited from outpatient clinic at the Freeman Hospital, United Kingdom. The intervention group will undergo telehealth behaviour intervention performed 3 times per week over 3 months. This program will comprise a home-based exercise (twice a week) and an individual lifestyle program (once per week). The control group will receive general health recommendations and advice to perform unsupervised walking training. The primary outcome will be feasibility and acceptability outcomes. The secondary outcomes will be objective and subjective function capacity, quality of life, dietary quality, physical activity levels, sleep pattern, alcohol and tobacco use, mental wellbeing, and patients' activation. This pilot study will provide preliminary evidence of the feasibility, acceptability and effectiveness of home-based telehealth exercise and behavioural intervention delivered in socioeconomically deprived patients with PAD. In addition, the variance of the key health outcomes of this pilot study will be used to inform the sample size calculation for a future fully powered, multicentre randomized clinical trial.

relevant data from this study will be made available upon study completion.

**Funding:** Funded by the League of Friends of Freeman Hospital and departmental funds. The funders had and will not have a role in study design, data collection and analysis, decision to publish, or preparation of the manuscript.

**Competing interests:** The authors have declared that no competing interests exist.

## Introduction

Peripheral arterial disease (PAD) results from chronic atherosclerosis that progressively leads to partial or total obstruction of the arteries, thereby, reducing blood flow and oxygen delivery to the peripheral regions of the body [1]. The main symptom of PAD is intermittent claudication, characterized by pain, cramp, or burning that occurs in the lower limbs during walking exercise and is relieved by rest [2]. Due to these symptoms, patients with PAD experience lower physical function [3], impaired cardiovascular function [4, 5] and lower quality of life [6]. Based on evidence demonstrating the benefits of exercise on walking capacity [7], cardiovascular function [5, 8] and quality of life [7], supervised exercise training is recommended in the United Kingdom by the National Institute for Health and Clinical Excellence as first-line treatment of claudication caused by PAD [9]. The National Institute for Health and Clinical Excellence also recommends that patients with PAD are offered advice, support, and treatment for the secondary prevention of cardiovascular disease, including smoking cessation, diet, and weight management [9].

Despite these recommendations, supervised exercise is rarely delivered in the clinical setting [10] and, patients are not systematically offered evidence-based interventions that support behavioural modification. Ineffectual provision of and, referral to, behaviour change programmes may contribute to further deterioration of health, leading many patients to undergo surgical procedures for PAD management (e.g., stent or bypass). While these procedures may provide short term benefits, they incur greater risk and cost [11, 12]. Also, without comprehensive behavioural intervention, patients are likely to resume the same poor health behaviours that contributed to PAD development initially. Therefore, there is imperative to develop, evaluate and integrate into care pathways 'prehabilitation' interventions that optimise patient health, prevent further deterioration, and improve long-term outcomes. Moreover, since lower socioeconomic status is associated with poor health status, multiple risk factors, unhealthy lifestyle, and more barriers to behaviour change programs [13, 14], this intervention program would most likely benefit socioeconomically deprived patients.

Usually, supervised exercise programmes and behavioural interventions are delivered face-to-face. However, the COVID-19 pandemic has significantly changed the organizational structure of health institutions and diverted attention to the pandemic management [15]. There have also been reports of outbreaks during exercise sessions, limiting face-to-face delivery of care to high-risk populations [16, 17]. Thus, home-based intervention programs delivered remotely (telehealth) are welcome as they represent an alternative option to manage PAD when inpatient or person-to-person rehabilitation is not possible. In general, telehealth is a cost-effective strategy [18, 19] and one of the greatest advantages is the possibility to deliver for patients who do not have access to traditional rehabilitation services (people who live in remote or deprived areas). In this sense, a telehealth program including home exercise training and lifestyle advice could have a significant impact on treatment outcomes and it would represent a 'COVID-proof', cost-effective and scalable care delivery option for these patients.

This pilot randomized controlled trial will establish the feasibility and acceptability of a 12-week home-based telehealth exercise and behavioural intervention delivered in socioeconomically deprived patients with PAD. The programme will be developed and delivered collaboratively between Newcastle upon Tyne NHS Trust, Northumbria University, Newcastle University, Healthworks and Newcastle United Foundation Club. The use of premier football team branding has been shown to improve the effectiveness of exercise and weight loss interventions and improve recruitment of 'hard to engage' men [20, 21]. This study will also determine the preliminary effectiveness of the intervention for improving clinical and health outcomes.

## Aims and objectives of the study

### Primary aim

To investigate the feasibility and acceptability of a 12-week home-based telehealth exercise and behavioural intervention in socioeconomically deprived patients with PAD.

### Primary objectives

To determine:

- Rates of patient screening, eligibility, recruitment, and retention to 12-week follow-up

- Patient compliance to the intervention (number of sessions attended and completed)

- Patient acceptability of the intervention through semi-structured qualitative interviews

### Secondary aims

To determine the preliminary effectiveness of a 12-week home-based telehealth exercise and behavioural intervention compared to usual care.

### Secondary objectives

To investigate whether a 12-week h home-based telehealth exercise and behavioural intervention compared to usual care:

- Improves functional capacity

- Reduces alcohol and tobacco use

- Improve diet quality

- Improves quality of life and mental wellbeing

- Increases daily ambulatory physical activity levels

### Tertiary aim

To explore the measurement of resource utilisation, costs and effects in an economic evaluation that would be conducted as part of a definitive randomized controlled trial.

### Tertiary objectives

To develop and test tools to measure the costs and effects of the health economic evaluation of 12-week home-based telehealth exercise and behavioural intervention compared to usual care.

## Materials and methods

### Study design

This is a single centred feasibility study and pilot randomised control trial assessing a 12-week home-based telehealth exercise and behavioural intervention in socioeconomically deprived patients with PAD. The Recommendations for Interventional Trials [22] (SPIRIT) flow chart and enrolment schedule, interventions and assessments for the trial are given in Fig 1.

|  | Enrolment | Allocation | Pre-evaluation | Intervention | Post-evaluation |
|---|---|---|---|---|---|
| **TIMEPOINT** | **-$t_1$** | **0** | **$t_1$** | **$t_2$** | **$t_3$** |
| **ENROLMENT:** |  |  |  |  |  |
| **Eligibility screen** | X |  |  |  |  |
| **Informed consent** | X |  |  |  |  |
| **Allocation** |  | X |  |  |  |
| **INTERVENTIONS:** |  |  |  |  |  |
| *Exercise and behavior change* |  |  |  | X |  |
| *Standard care* |  |  |  | X |  |
| **ASSESSMENTS:** |  |  |  |  |  |
| *Feasibility outcomes* |  |  |  |  | X |
| *Objective function capacity* |  |  | X |  | X |
| *Subjective function capacity* |  |  | X |  | X |
| *Quality of Life* |  |  | X |  | X |
| *Dietary quality* |  |  | X |  | X |
| *Physical activity levels* |  |  | X |  | X |
| *Sleep* |  |  | X |  | X |
| *Alchool and tobacco use* |  |  | X |  | X |
| *Mental wellbeing* |  |  | X |  | X |
| *Patient activation* |  |  | X |  | X |
| *Resource utilization* |  |  | X |  | X |

**Fig 1. The recommendation of interventional trials (SPIRIT) schedule of enrolment, interventions, and assessments.**

## Ethical approval and consent to participate

The protocol study was approved by the Research Ethics Committee of Human Research of Freeman Hospital (April 2021—IRAS Reference Number: 286735), registered and published in the ClinicalTrial.gov (registration number: NCT05260567) and will be conducted according to the principles of the Declaration of Helsinki. Patients will be provided with appropriate participant information sheets designed in compliance with national guidance. They will have adequate time to consider the information, ask questions and have them answered sufficiently. Patients will be advised that participation is voluntary and that they may withdraw from the

study at any time without having to provide a reason or affecting their care. Patients who are willing to participate will be asked to sign a consent form, and this process will be conducted by a trained delegated member of the research team. A copy of the signed consent form and participant information sheet will be filed in the patient's medical notes and a further copy will be given to the patient. The original signed consent form will be retained in the Investigator site file and the patients' general practiser will be informed of their participation in the study.

## Eligible participants

Patients attending the Vascular Unit of Freeman Hospital–Newcastle Upon Tyne—UK with a diagnosis of PAD will be assessed for potential suitability by the clinical team and then contacted by the research team by telephone after their clinic appointment/further investigations. Patient eligibility will be based on the criteria below.

## Inclusion criteria

- Diagnosis of PAD confirmed by ankle brachial index <0.90 in one or both limbs
- Age > = 40 years
- Able to walk distance >50m
- Live in an area deemed in lowest 30% of super output area from Office of National Statistics

## Exclusion criteria

- chronic limb threatening ischemia
- short claudication distance <50m
- severe heart disease (Grade III or IV, New York Heart Association)
- severe ischemic or haemorrhagic stroke or neurodegenerative diseases
- severe hypertension (systolic blood pressure of more than 180 mm Hg, and diastolic blood pressure of more than 100 mm Hg)
- uncontrolled cardiac disease (presence of complex arrhythmias, unstable angina during the previous month and myocardial infarction during the previous month)
- a resting heart rate of more than 120 beats per minute
- has already undergone angioplasty, bypass or other surgical intervention for PAD
- other severe comorbid conditions preventing the ability to engage in physical activity inability or unwillingness to undertake the commitments of the study

## Randomisation

Patients will be randomised in a 1:1 ratio to either the home-based telehealth exercise and behavioural intervention or standard care. The randomization will be performed in blocks of 15 patients, using an online randomisation generator (www.randomizer.org).

## Interventions

The intervention has been co-designed with Newcastle upon Tyne NHS Trust, Northumbria University, Healthworks and the Newcastle United Foundation. Patients in the intervention group will receive educational materials and videos outlining the main intervention components and how health behaviours impact upon their condition. The videos will be Newcastle United Foundation Club/Healthworks branded and be developed by the individuals from these organizations. Patients will also receive a Newcastle United Foundation Club T-shirt. Shortly after allocation to the intervention group, patients will be contacted by a Health Trainer from Healthworks who will conduct an initial assessment and consultation. Health trainers have various qualifications in health care e.g., nutrition degree, Level 4 rehabilitation qualification, all receive Healthworks training in motivational interviewing and intervention delivery. Patients will meet with their dedicated health trainer weekly for one hour via phone call/videoconference for 12 weeks and discuss the behaviours outlined below. Health trainers use many behaviour change techniques to promote modification of risk factors such as goal setting, problem solving and self-regulation.

## Smoking cessation

Self-reported smoking habits will be assessed at baseline. Patients who smoke will receive a cessation intervention from the health trainer i.e., discussion of previous quit attempts and benefits of quitting to aid in improving health and exercise capacity. If required, nicotine replacement therapy vouchers redeemable at their local pharmacy will be posted to the patient. An eight-week supply of e-cigarette cartridges may also supply.

## Alcohol intervention

Health trainers will deliver a previously evaluated brief behavioural intervention to reduce alcohol intake to low-risk levels (<14 units per week) [23]. Intervention materials incorporate specific techniques that target intention formation and enactment of behaviour change (e.g. information on health consequences, social support, goal setting behaviour, problem solving, restructuring the physical environment). Patients suspected to have an alcohol use disorder or risky drinking at baseline will receive additional intervention from their general practitioner.

## Nutrition

Patients will receive basic nutrition education and health eating advice in line with recommendations from the British Heart Foundation and Diabetes UK. Health Trainers will provide help to overcome barriers to healthy eating.

## Mental well-being and finances

Patients will receive a light-touch intervention on sleep hygiene and stress management. If we discover that the patient is severely depressed, has self-harming concerns or suicidal thoughts the GP will be contacted, or patient signposted for further help. Health trainers will also help patients to determine their eligibility to receive benefits and, facilitate access and uptake if required.

## Supervised exercise

The home-based exercise training will be performed twice a week for 12 weeks via Zoom (up to 5 patients per session). Each session will be comprised of warm-up (10 min), the main part (15 to 20 min), and cooldown (5 to 10 min). The training aims to develop resistance, aerobic

and functional capacity such as getting up, walking, pulling, pushing, throwing, and transferring body weight or external loads. An example of multimodal exercise training is shown in Box 1.

### Box 1. Example of exercise training module in the intervention

| Duration | Category | Exercise | Intensity (Borg Scale) |
|---|---|---|---|
| 10 min | Warm-up | • Active and dynamic joint mobility; coordination, balance, displacement, spatial orientation and proprioception exercises. | Very light to fairly light |
| 15 to 20 min | Resistance | • Resistance exercise for upper and lower limbs<br>• 6 to 8 exercises<br>• 2–3 sets of 8 to 10 repetitions<br>• Interval sets 1'30" a 2 min | Somewhat hard to hard |
| | Flexibility | • Emphasis on joint mobility exercises,<br>• Maintenance of static positions combined with breathing techniques;<br>• Proprioceptive neural facilitation techniques;<br>• 40 sec to 1 min each exercise | Somewhat hard to hard |
| | Aerobic exercise (circuit training) | • Global exercises, involving large muscle groups focused on aerobic capacity.<br>• Circuit of 3 to 4 exercises<br>• Stimulus– 30 sec to 1 min<br>• Passive interval (1 min) | Somewhat hard to hard |
| 5 to 10 min | Cooldown | • Active and static stretching exercises;<br>• Breathing relaxation exercises; | Very light to fairly light |

The training intensity will be progressively adjusted by increasing the load (e.g., using common household objects), increasing the complexity, speed of movements and volume of exercises by varying circuits. The intensity of the exercise will be monitored using the Borg scale (from 0 to 20) with target intensity zone from 12 to 14 (Somewhat hard to hard) [24].

In addition to the home-based training sessions, patients will be encouraged to increase their physical activity. Patients will be provided with a Fitbit device to monitor their step count and will be recommended to increase their previous week's average step count by 10%. Participants who have access to the internet will be asked to upload data to the Fitabase research platform. Data will be anonymised and will only be accessible to the research number.

## Usual care

Patients randomized to the standard care group will receive general recommendations (Box 2) to modify risk factors and standard care as per trust guidelines given in their routine outpatient clinic appointment. Patients will also receive specific advice to perform unsupervised walking exercise for around 30 minutes three to five times a week, according to recently published NICE guidelines [9]. Patients will also receive a Fitbit device so they can measure their own exercise capacity and increase as recommended. They will be asked to upload their data to the research platform, if possible, as per the lifestyle intervention group.

> ## Box 2. Walking intervention for patients randomised to standard care
>
> **Step 1**: Warm up. Stretch your calf and thigh muscles in each leg for 10 to 15 seconds.
>
> **Step 2**: Start walking. Walk at a fast-enough pace for about 5 minutes, even though it may cause some mild pain.
>
> **Step 3**: Stop and rest. After 5 minutes of mild or moderate pain, stop and rest until the pain goes away.
>
> **Step 4**: Repeat the walk-and-stop routine several times. During the first two months of your walking program, build up slowly to walking a total of 35 minutes each session, not counting the rest breaks. Keep adding a few minutes until you're at the goal of walking 50 minutes.
>
> **Step 5**: Cool down. Finish by walking slowly for 5 minutes. Then, stretch your calf and thigh muscles again.
>
> **Step 6**: Stick with it.

## Outcomes measurements

**Primary outcomes.** Feasibility will be determined by calculating the rate of patient screening, eligibility, recruitment, retention at 12 weeks and adherence to the intervention (number of sessions attended and completed). Patient acceptability of the intervention and study experience more broadly will be determined through semi-structured qualitative 1-2-1 interviews and/or focus groups. Given the challenges of conducting focus groups remotely, and anticipated characteristics of the participant group, we expect that 1-2-1 interviews will be more practical and facilitative. We will purposively sample participants from both study arms so that the experiences of patients from varied ethnic backgrounds, age and gender are represented. Acceptability of the intervention will be guided by Sekhon et al.'s Theoretical Framework of Acceptability [25] and the NIH Behavior Change Consortium's Best Practices and Recommendations. We will also ask patients about their experience of participating in the study e.g., informed consent process, time commitment, acceptability of measures e.g. readability, burden etc. We will also ask for their experiences, thoughts, and attitudes towards current usual care for PAD. Interviews will be conducted until data saturation is reached in a maximum of 20 participants. All focus groups will be audio recorded and transcribed, and these data will be analysed thematically to generate themes and outcomes.

**Secondary outcomes.** *Objective functional capacity*. Patients will complete the 6-minute walk test (6MWT) [26] at baseline and 12-week follow-up. Briefly, patients will be encouraged to "walk at their usual pace for six minutes and cover as much ground as possible" and rest if necessary. The outcomes will be the onset claudication distance (distance walked when the patients related the occurrence of symptoms of intermittent claudication [27]) and six-minute total walking distance (6MWD; the maximum distance achieved by the patient at the end of the test). The test will be administered by a trained member of staff and conducted in line with the American Thoracic Society guidance [26].

*Subjective functional capacity*. Patients will complete the Walking Impairment Questionnaire [28] at baseline and 12-week follow-up to assess three factors of walking impairment: walking distance, walking speed, and the ability to climb stairs. Patients will be asked how difficult it was to walk in these situations should answer as "none, slight, some, much or unable". Each domain is anchored from 0, representing extreme limitation, to 100 representing no difficulties. Patients will also complete the Walking Estimated Limitation Calculated by History questionnaire [29] at both time points. Patients report how long they can walk at certain speeds, and then how they would rate their speed of walking relative to their relatives, friends, or people at same age.

## Quality of life

Patients will complete the vascular quality of life questionnaire [30] at baseline and 12-week follow-up. The measure is composed of six items evaluating the impact of vascular disease on social aspects and capacity to perform daily activities. Each item is scored 1–4. The total score is achieved by summarizing the score on each item, resulting in a score between 6 and 24. Higher value indicates better health status. Patients will also complete the EuroQoL questionnaire (EQ-5D-5L) [34] at both study time points which measures five dimensions: mobility, self-care, usual activities, pain/discomfort and anxiety/depression [31]. The digits for the five dimensions can be combined into a 5-digit number that describes the patient's health state [32].

## Dietary quality

Participants will complete the Short Form Dietary Questionnaire which comprises 24 items that collect data on dietary intake frequency. Participants' responses will be used to derive a dietary quality score. The tool is shown to be a valid method of assessing dietary quality in UK adults [33].

## Physical activity levels

Daily ambulatory activity will be assessed using a wrist-worn accelerometer (Fitbit Charge HR) which measures step count, resting heart rate and time in sedentary behaviour as well as light, moderate and vigorous activities. Fitbit devices have been shown to have good concurrent validity for measuring sedentary behaviour and physical activity compared to research-grade accelerometers [34–36]. Patients will be instructed to wear the device every day during the study and asked to charge the device every 48–36 hours overnight. Patients' anonymised activity data will be uploaded to the Fitbit online dashboard (Fitabase) and extracted by researchers for analyses.

## Sleep

Sleep quality and quantity will also be recorded by the Fitbit device. Patients' anonymised sleep data will be uploaded to Fitabase and extracted by researchers for analyses. Wrist-worn Fitbit devices are shown to have good validity for obtaining gross estimates of sleep parameters [37].

## Alcohol and tobacco use

The 3-item Alcohol Use Disorders Identification Test—Consumption (AUDIT-C) screening tool [38] will be administered to patients at baseline and 12-week follow-up to identify alcohol use disorders or risky drinking. The tool has good validity in diverse populations [39, 40]. If the patient is suspected to have an alcohol use disorder or risky drinking based on their AUDIT C score, their general practitioner will be contacted to offer further input and advice. Patients will complete a standard 30 second carbon monoxide breath test at baseline and follow- up to formally assess smoking status. This is not an aerosol generating procedure and therefore will be appropriate to use. Self-reported smoking habits will be assessed at baseline and follow-up. Patients will be asked to report their current tobacco smoking status and frequency of smoking cigarettes/other tobacco products and, previous tobacco smoking status.

### Mental wellbeing

Patients will complete the 14-item hospital anxiety-depression score [41] at baseline and 12-week follow-up. The 14-item instrument is comprised of a depression subscale and anxiety subscale which are each assessed through 7-items. Respondents will be asked to rate their mental and emotional state over the past week. The hospital anxiety-depression score has good validity in community and clinical populations [41, 42].

### Patient activation

The Patient Activation Measure (PAM®) [43] measures patients' knowledge, skills and confidence in managing their condition. PAM licences are available from NHS England and Improvement as part of the Supported Self-management component of the Personalised Care Programme. At baseline and 12-week follow-up patients will respond (strongly disagree/disagree/agree/strongly agree/N/A) to 13 statements related to their confidence in managing their health. Patients' PAM score (0–100) will place them within one of four activation categories, providing insight into a range of health-related characteristics and behaviours. The PAM has been validated for use in adults with long term conditions [44–46].

### Resource utilisation

The case report forms will measure the resource utilisations by each patient. The case report forms will be administered to each patient at baseline and 12-week follow-up in both arms of the trial asking them to report any health care resources they have utilised in the preceding 12 weeks.

### Health economics

The trial will not be adequately powered to conduct a formal health economic evaluation, therefore the data accrued from the pilot trial will be reported using descriptive statistics to explore the differences in resource utilisations, costs and effects between the trial arms. We will also assess the completion rates for the health economics data collection tools, responses for each question and health state utility values. At baseline and 12-week follow-up, all patients will complete the following: EuroQol's EQ-5D-5L, EQ-5D VAS questionnaire and case report forms measuring health resource utilisation.

### Study power

As per good practice recommendations for pilot studies [47], we will aim to recruit 30 patients to each arm and obtain a total sample of 60 participants. The findings of the current study will be used to inform the power calculations of a future definitive randomised controlled trial.

### Statistical plan

Primary outcome data will be reported as descriptive statistics, including rates of: patient screening, eligibility, recruitment and retention to 12-week follow-up and survey completion. Qualitative interview data will be analysed thematically to generate themes. Exploratory between-group analyses will be conducted to determine preliminary intervention effectiveness. Normality and homogeneity of variance will be performed using the Shapiro-Wilks and Levene tests, respectively. For comparison of the variables at the pre-intervention, we will perform a one-way ANOVA. To analyze the responses before and after the intervention period, two-way analysis of variance for repeated measures will be used, with the main factors being

the group (and the time (pre and post-intervention) with Newman-Keuls post-hoc test. The level of significance will be set at P <0.05.

## Trial status

Enrolment of patients started in February 2022.

## Discussion

To the best of our knowledge, this is the first study to assess the feasibility and acceptability of a 12-weeks home-based telehealth exercise and behavioural intervention delivered in patients with PAD living in socioeconomically deprived areas. In addition, this study will also determine the preliminary effectiveness of the intervention for improving clinical and health outcomes in these patients.

Supervised exercise training and secondary cardiovascular prevention are considered a cornerstone for clinical treatment in patients with PAD due to the benefits in different health parameters, such as functional capacity, cardiovascular function, and quality of life [5, 7, 48, 49]. Despite these benefits, in the UK patients has limited access to supervised exercise training in the clinical setting and, patients are not systematically offered evidence-based interventions that support behavioural modification for their risk factors [10]. In addition, a recent study found that financial limitations and travel distances were cited as barriers to participation in supervised exercise training in PAD patients in the UK. [10]. To tackle these barriers, we will test the feasibility, acceptability and effectiveness of a home-based exercise and behavioural intervention in patients with PAD. This program will be delivered remotely to patients living in socioeconomically deprived areas in the North East of England, known as one of the highest prevalence of unhealthy habits in the UK [50].

If effective, we can support preliminary evidence that our home-based telehealth behavioural intervention may be a feasible strategy for PAD treatment and can be incorporated into public services deliveries. In addition, the variance of the key health outcomes of this pilot study will be used to inform the sample size calculation for a future definitive multicentre randomized clinical trial.

## Supporting information

**S1 Checklist. SPIRIT 2013 checklist: Recommended items to address in a clinical trial protocol and related documents.**
(DOC)

**S1 File.**
(PDF)

## Author Contributions

**Conceptualization:** Gabriel Cucato, Chris Snowden, Emma McCone, Craig Nesbitt, Sandip Nandhra, Mackenzie Fong, Eileen Kane, Maisie Rowland, Nawaraj Bhattarai, Paul Court, Oliver Bell, John Michael Saxton, James Prentis.

**Funding acquisition:** James Prentis.

**Methodology:** Gabriel Cucato, Craig Nesbitt, Sandip Nandhra, Maisie Rowland, Paul Court, Oliver Bell, John Michael Saxton, James Prentis.

**Project administration:** Emma McCone, Mackenzie Fong.

**Writing – original draft:** Gabriel Cucato, Chris Snowden, Emma McCone, Mackenzie Fong, Eileen Kane, John Michael Saxton, James Prentis.

**Writing – review & editing:** Gabriel Cucato, Chris Snowden, Emma McCone, John Michael Saxton, James Prentis.

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
