## [Decision Letter · Decision Letter 0]

27 Apr 2022

PONE-D-22-06371Evaluating the feasibility and acceptability of an exercise and behaviour change intervention in socioeconomically deprived patients with peripheral arterial disease: THE TEXTPAD STUDYPLOS ONE

Dear Dr. Cucato,

Thank you for submitting your manuscript to PLOS ONE. After careful consideration, we feel that it has merit but does not fully meet PLOS ONE’s publication criteria as it currently stands. Therefore, we invite you to submit a revised version of the manuscript that addresses the points raised during the review process.

We look forward to receiving your revised manuscript.

Kind regards,

Yoshihiro Fukumoto

Academic Editor

PLOS ONE

Journal Requirements:

2. During your revisions, please note that a simple title correction is required: As this is a study protocol we require the word protocol to be in the title . Please ensure this is updated in the manuscript file and the online submission information.

4. Please ensure that you refer to Figure 2 in your text as, if accepted, production will need this reference to link the reader to the figure.

5. Please upload a copy of Figure 2, to which you refer in your text on page 30. If the figure is no longer to be included as part of the submission please remove all reference to it within the text.

Reviewers' comments:

Reviewer's Responses to Questions

**Comments to the Author**

1. Does the manuscript provide a valid rationale for the proposed study, with clearly identified and justified research questions?

Reviewer #1: Yes

Reviewer #2: Partly

2. Is the protocol technically sound and planned in a manner that will lead to a meaningful outcome and allow testing the stated hypotheses?

Reviewer #1: Yes

Reviewer #2: Partly

3. Is the methodology feasible and described in sufficient detail to allow the work to be replicable?

Reviewer #1: Yes

Reviewer #2: No

4. Have the authors described where all data underlying the findings will be made available when the study is complete?

Reviewer #1: No

Reviewer #2: No

5. Is the manuscript presented in an intelligible fashion and written in standard English?

Reviewer #1: Yes

Reviewer #2: No

6. Review Comments to the Author

You may also provide optional suggestions and comments to authors that they might find helpful in planning their study.

Reviewer #1: This is a protocol paper investigating about the feasibility and acceptability of an exercise and behavior change intervention in patients with peripheral arterial disease (PAD). This research focused on patients with PAD in socioeconomic deprived area. The protocol was arranged well in detail. There were some points to be addressed.

Introduction

# What is the reason for the limitation of study patients to socioeconomically deprived area? More concise explanation about it should be added.

Method

# The issue about “uncontrolled cardiac arrhythmias” were hard to understand.

What is the association between “uncontrolled cardiac arrhythmias” and “unstable angina or myocardial infarction”, which was mentioned in parentheses?

#This study uses telehealth devise such as zoom application or fitbit device. How about the preparation of these devices? In addition, patients with low cognitive function might have a hard time handling these devices. Please comment about how to handle the disparity in the availability for IT devices.

Discussion

# Authors described the barrier of cardiac rehabilitation in PAD. More concise documentation for the barrier should be discussed. Indeed, there are several levels of the barriers (patient level and physician level) (Expert Rev Cardiovasc Ther. 2020 Nov;18(11):777-789). In addition, covid infection worsened the difficulty in delivering cardiac rehabilitation to each patient.

Reviewer #2: In the present manuscript submitted by Cucato et al., there are typos and wrong line spaces a lot, and the reference list is lacking. It looks like a draft before submission. Was the present manuscript reviewed by all authors before submission? I feel that the authors should format the description in whole.

7. PLOS authors have the option to publish the peer review history of their article (what does this mean?). If published, this will include your full peer review and any attached files.

Reviewer #1: No

Reviewer #2: No

---

## [Author Response · Author response to Decision Letter 0]

26 May 2022

Please find enclosed the new version of the manuscript: “Evaluating the feasibility and acceptability of an exercise and behaviour change intervention in socioeconomically deprived patients with peripheral arterial disease: THE TEXTPAD STUDY PROTOCOL”.

We have revised the manuscript according to the reviewer’s comments and have included the change within the manuscript which is highlighted in yellow. We hope you find this paper acceptable for publication. Thank you for your time and consideration.

Sincerely,

 

Reviewer #1: This is a protocol paper investigating about the feasibility and acceptability of an exercise and behavior change intervention in patients with peripheral arterial disease (PAD). This research focused on patients with PAD in socioeconomic deprived area. The protocol was arranged well in detail. There were some points to be addressed.

1) Introduction

# What is the reason for the limitation of study patients to socioeconomically deprived area? More concise explanation about it should be added.

Answer: We only included patients in socioeconomically deprived areas because they present poor health status, presence of multiple risk factors, unhealthy lifestyle and more barriers to behaviour change programs [1, 2] when compared to patients living in affluent areas. We wanted to ensure that this type of intervention is feasible and that it promotes preliminary evidence of effectiveness on overall health in deprived areas before we appropriate it for all patients with PAD regardless of socioeconomic status.

2)Method

# The issue about “uncontrolled cardiac arrhythmias” were hard to understand.

What is the association between “uncontrolled cardiac arrhythmias” and “unstable angina or myocardial infarction”, which was mentioned in parentheses?

Answer: We would like to be sorry about this mistake. We correct this error in the new version of the manuscript. 

3) #This study uses telehealth devise such as zoom application or fitbit device. How about the preparation of these devices? In addition, patients with low cognitive function might have a hard time handling these devices. Please comment about how to handle the disparity in the availability for IT devices.

Answer: On the day that patients are recruited, our team provides all information regarding how to use the smartwatch (Fitbit) and how to access classes via zoom. In addition, HealthWorks offers telephone support for patients experiencing difficulties logging into classes. There have been only a few issues with exercise class access, which our team has quickly resolved.

4) Discussion

# Authors described the barrier of cardiac rehabilitation in PAD. More concise documentation for the barrier should be discussed. Indeed, there are several levels of the barriers (patient level and physician level) (Expert Rev Cardiovasc Ther. 2020 Nov;18(11):777-789). In addition, covid infection worsened the difficulty in delivering cardiac rehabilitation to each patient.

Answer: As recommended, we discussed in more detail the possible barrier to exercise training in PAD patients. 

Reviewer #2: 

1) In the present manuscript submitted by Cucato et al., there are typos and wrong line spaces a lot, and the reference list is lacking. It looks like a draft before submission. Was the present manuscript reviewed by all authors before submission? I feel that the authors should format the description in whole.

Answer: We would like to thank you for your comments. All authors approved the final version of the manuscript. However, we made a new author’s review and updated references and manuscript formatting. 

REFERENCES

1. Foster HME, Celis-Morales CA, Nicholl BI, Petermann-Rocha F, Pell JP, Gill JMR, et al. The effect of socioeconomic deprivation on the association between an extended measurement of unhealthy lifestyle factors and health outcomes: a prospective analysis of the UK Biobank cohort. Lancet Public Health. 2018;3(12):E576-E85. doi: 10.1016/S2468-2667(18)30200-7. PubMed PMID: WOS:000452029600011.

2. Michie S, Jochelson K, Markham WA, Bridle C. Low-income groups and behaviour change interventions: a review of intervention content, effectiveness and theoretical frameworks. J Epidemiol Commun H. 2009;63(8):610-22. doi: 10.1136/jech.2008.078725. PubMed PMID: WOS:000267941400007.

---

## [Decision Letter · Decision Letter 1]

2 Jun 2022

Evaluating the feasibility and acceptability of an exercise and behaviour change intervention in socioeconomically deprived patients with peripheral arterial disease: THE TEXTPAD STUDY PROTOCOL

PONE-D-22-06371R1

Dear Dr. Cucato,

We’re pleased to inform you that your manuscript has been judged scientifically suitable for publication and will be formally accepted for publication once it meets all outstanding technical requirements.

Kind regards,

Yoshihiro Fukumoto

Academic Editor

PLOS ONE

Additional Editor Comments (optional):

Reviewers' comments:

Reviewer's Responses to Questions

**Comments to the Author**

1. Does the manuscript provide a valid rationale for the proposed study, with clearly identified and justified research questions?

Reviewer #1: Yes

Reviewer #2: Yes

2. Is the protocol technically sound and planned in a manner that will lead to a meaningful outcome and allow testing the stated hypotheses?

Reviewer #1: Yes

Reviewer #2: Yes

3. Is the methodology feasible and described in sufficient detail to allow the work to be replicable?

Reviewer #1: Yes

Reviewer #2: Yes

4. Have the authors described where all data underlying the findings will be made available when the study is complete?

Reviewer #1: Yes

Reviewer #2: Yes

5. Is the manuscript presented in an intelligible fashion and written in standard English?

Reviewer #1: Yes

Reviewer #2: Yes

6. Review Comments to the Author

You may also provide optional suggestions and comments to authors that they might find helpful in planning their study.

Reviewer #1: Thank the authors for the correcting the previous version of the manuscript. The revised manuscript was finely corrected.

Reviewer #2: I have read the revised manuscript, and I have no comment for the revised text, tables (boxes), and figure.

7. PLOS authors have the option to publish the peer review history of their article (what does this mean?). If published, this will include your full peer review and any attached files.

Reviewer #1: **Yes: **Eisuke Amiya

Reviewer #2: No

---

## [Editor Report · Acceptance letter]

14 Jun 2022

PONE-D-22-06371R1 

Evaluating the feasibility and acceptability of an exercise and behaviour change intervention in socioeconomically deprived patients with peripheral arterial disease: THE TEXTPAD STUDY PROTOCOL 

Dear Dr. Cucato:

I'm pleased to inform you that your manuscript has been deemed suitable for publication in PLOS ONE. Congratulations! Your manuscript is now with our production department. 

Kind regards, 

on behalf of

Dr. Yoshihiro Fukumoto 

Academic Editor

PLOS ONE